# A 10-Year Impact Evaluation of the Universal Salt Iodization (USI) Intervention in Sarawak, Malaysia, 2008–2018

**DOI:** 10.3390/nu14081585

**Published:** 2022-04-11

**Authors:** Lim Kuang Kuay, Noor Ani Ahmad, Tan Beng Chin, Chan Ying Ying, Maznieda Mahjom, S. Maria Awaluddin, Noor Syaqilah Shawaluddin, Tuan Mohd Amin Tuan Lah, Tahir Aris

**Affiliations:** 1Institute for Public Health, National Institutes of Health, Ministry of Health Malaysia, Shah Alam 40170, Malaysia; drnoorani@moh.gov.my (N.A.A.); chan.yy@moh.gov.my (C.Y.Y.); drmaznieda@moh.gov.my (M.M.); smaria@moh.gov.my (S.M.A.); noorsyaqilah@moh.gov.my (N.S.S.); tuanmohdamin@moh.gov.my (T.M.A.T.L.); 2Sarawak State Health Department, Ministry of Health Malaysia, Kuching 93050, Malaysia; tanbc@moh.gov.my; 3Institute for Medical Research, National Institutes of Health, Ministry of Health, Kuala Lumpur 50588, Malaysia; tahir.a@moh.gov.my

**Keywords:** USI, urinary iodine concentration, goitre, school children, Sarawak

## Abstract

A universal salt iodization (USI) was introduced in Sarawak, Malaysia in 2008 to control the iodine deficiency disorders (IDD) among its population. The purpose of this study is to evaluate the impact of the USI among school children in Sarawak after 10 years of implementation. The data were extracted from 2008 and 2018 Sarawak state-wide IDD surveys. Briefly, both surveys were cross-sectional surveys covering information on the socio-demographic, status of goitre, urinary iodine, and the amount of iodine in the salt samples. A total of 1104 and 988 between the ages of 8 and 10 were involved in the 2008 and 2018 surveys, respectively. The overall prevalence of goitre among the school children in Sarawak was significantly lower in 2018 (0.1%) compared to 2008 (2.9%). The median urinary iodine concentration (UIC) in urine samples had risen significantly from 102.1 µg/L in 2008 to 126.0 µg/L in 2018. In terms of iodine content in salt samples, the median concentration improved significantly in 2018 (35.5 µg/L) compared to 2008 (14.7 µg/L). After 10 years of USI implementation in Sarawak, the results from both surveys confirmed the effectiveness of mandatory USI in increasing the nutritional iodine status of school children in Sarawak.

## 1. Introduction

Iodine is a mineral that is necessary for normal growth and development of the fetus, infant, child, as well as for adults’ normal physical and mental activity [1]. It presents in soil and is imbibed through foods grown in the soil. However, in many areas, repeated floods and glacier erosion have leached iodine from the surface soil [2]. Thus, animal and human population which are completely reliant on food produced in such soil are unable to obtain the physiological amount of iodine and deficiency symptoms appear, leading in a slew of developmental and functional issues known as iodine deficiency disorders or IDD [3]. Over the last decade, global iodine nutrition has significantly improved, and the number of iodine-deficient nations has significantly reduced. However, about two billion people worldwide, including 241 million children, still have inadequate dietary iodine intakes [4,5]. 

The simplest technique to ensure optimal iodine nutrition is through universal salt iodization (USI) [6]. There are few reasons why iodized salt is the preferred mode for IDD prevention and control: the vast majority of people consume salt as a common part of a meal, amounts consumed are relatively constant, and sources of salt are usually limited making the addition of iodine a feasible option [7]. Taking into account several factors: iodine loss from salt is 20% from production site to household, and 20% more during cooking, and average salt intake per capita of 10 g/day, which result in a final estimation of 23 mg I/kg. Therefore, the content of 40 mg I/kg is applicable when the salt intake is slightly higher than 5 g/day [8]. 

In Sarawak, a large-scale survey in 1970 and large number of focal surveys in 1980s showed the prevalence of goitre in endemic areas ranged from 0.7 to 99.5% among the communities primarily in the state’s interior areas [9,10,11]. It is not surprising to observe the high prevalence of goitre among the community in the rural Sarawak due to a less access of iodine-rich seafoods and a high intake of goitrogens, such as cassava roots and leaves. However, the level of socio-economic development that has taken place, particularly during the last 20 years, have made outside food, including seafoods, more widely available to communities living in once remote and inaccessible areas [12,13]. In 1982, Sarawak’s State Health Department passed laws designating 19 districts and three sub-districts as “Goitre Endemic Areas” in rural areas, only iodized salt is permitted for sale. However, due to a lack of legislative enforcement, IDD continued to be a significant public health issue particularly in the less developed, difficult to access interior [12]. In the 1990s, a new short-to-medium-term approach to iodine delivery was adopted by supplementing iodized water to villages and schools/hostels where IDD is known to exist. The locally modified iodizing system offers a new cost-effective strategy for the control of endemic IDD in all iodine-deficient areas in Sarawak [13]. This innovation iodinated water supply equipment (iodinator) had successfully reduced the goitre prevalence by as high as 35% and increased the median urinary excretions to levels recommended for iodine sufficient population in many parts of the state [12]. 

According to a national IDD survey performed in Malaysia in 2008, the median urinary iodine concentration (UIC) of school children in Sarawak was 101.9 g/L, while the median UIC of children from rural regions was 96.6 g/L [14]. Following the outcome of the 2008 IDD survey, a decision was made by Malaysia’s Ministry of Health to introduce universal salt iodization (USI) in the state. In July 2008, legislation was passed requiring all salt marketed for human consumption in the state to contain between 20 mg/kg to 40 mg/kg of iodide calculated as potassium iodide [15]. This paper compares the results from two state-wide IDD surveys conducted in 2008 and 2018 with the purpose to assess the impact of the USI on Sarawak’s school children after ten years of implementation.

## 2. Materials and Methods

### 2.1. Study Sample

Data of the Iodine Deficiency Disorder surveys were obtained from both state-wide IDD surveys in 2008 and 2018. To improve the chances of our outcomes being comparable, the school-based cross-sectional study conducted in 2018 used similar approach as the previous survey conducted in 2008. Briefly, the WHO/UNICEF/ICCIDD recommended 30 clusters sampling methodology was utilised to choose schools and students [8]. All primary schools in Sarawak under the Ministry of Education were used as sampling units for the survey. Sarawak State Education Department provided a list of schools and their student enrolment. Subsequently, using the proportionate to population size (PPS) sampling technique, a total of 30 schools were chosen. Systematic sampling was used to randomly select 40 children aged 8 to 10 years old from each of the 30 schools. A total of 1200 school children took part in the surveys in year 2008 and 2018, respectively. 

### 2.2. Assessment of Thyroid Volume

The visual examination and palpation were used to determine the thyroid status, where the medical staff nurse stood behind the seated children, with two thumbs placed several centimetres below the notch of the thyroid cartilage on either side of the windpipe and rolling the thumbs over the thyroid next to the windpipe. The WHO/UNICEF/ICCIDD criteria were used to categorise goitre, where grade 0 indicates no palpable or visible goitre when the neck is in the normal position, grade 1 indicates a goitre that is palpable but not visible, and grade 2 indicates a swelling in the neck that is clearly visible when the neck is in the normal position and is consistent with an enlarged thyroid when the neck is palpated. The total goitre rate (TGR) was used to determine the severity of IDD; 5.0–19.9% was considered mild, 20.0–29.9% was considered moderate, and 30.0% or more was considered severe [8]. 

### 2.3. Urinary Iodine Determination

Each child had a spot urine sample taken by using a tightly closed 20 mL plastic urine bottle. The samples were stored in the freezer at the district health office, and frozen before courier to the regional IDD laboratory in Sabah, Malaysia. Urinary iodine levels were determined using an in-house customised micro-method based on manual ammonium persulfate digestion followed by calorimetric determination of the Sandel–Kolthoff reaction. When a reading exceeded the reference standard, the sample was diluted and reanalysed [16]. The school children with UICs of <20, 20–49, 50–99 were classified as severe, moderate, and mild, respectively, while the UICs of 100–199 μg/L, 200–299 μg/L, and ≥300 μg/L were classified as adequate, above requirement, and excessive, respectively [8]. In addition of optimal iodine status indication, less than 20% of the population samples should have UIC < 50 µg/L [8]. According to a study on thyroid function and iodine status, the median UIC range of 100–299 μg/L was not related with any thyroid dysfunction [17]. Thus, the median UIC in school children with 100–199 μg/L and 200–299 μg/L will be presented as a single group in the current analysis [18].

### 2.4. Salt Iodine Determination

Every school child was requested to bring 80 g of table salt from home in self-sealing plastic bags. Subsequently, 10% of the total number of salt samples brought by students were chosen at random for iodine level determination using the iodometric titration method [19] at the IDD national reference laboratory, Institute for Medical Research, Kuala Lumpur. All the samples were analysed in duplicates and the results were expressed as mg/kg. Based on the current Malaysian salt regulation, the proportion of iodine content in the salt was divided into three categories: <20 mg/kg was considered as non-satisfactory, 20–40 mg/kg as satisfactory and >40 mg/kg as excess [15]. 

### 2.5. Data Analysis

SPSS version 25 was used to conduct all statistical analyses (IBM SPSS, Chicago, IL, USA). Descriptive statistics (mean, standard deviation, medians, interquartile range, kurtosis, and skewness), and the Kolmogorov–Smirnov test was used to assess normality. The data of the urinary were not normally distributed. Thus, the median and interquartile range were used to present continuous values, while frequency and proportion were used to present categorical variables. The Pearson Chi-square test was used to compare differences in goitre prevalence between populations, while the Wilcoxon-Mann–Whitney test was used to determine differences in median UIC between groups. 0.05 was used as the statistical significance level.

## 3. Results

A total of 1104 and 988 school children participated in the IDD survey in 2008 and 2018, with the response rate of 92.0% and 82.3%, respectively. Out of 1104 children participated in the 2008 study, more than half of the respondents (51.5%) were from Southern division, followed by Central and Southern division. Out of 988 respondents participated in the 2018 study, the highest proportion was from Southern division (35.6%). In both studies, the school children were equally distributed by age and gender (Table 1).

In 2008, the total goitre rate (TGR) identified was only 2.9% (n = 32), indicating Sarawak was not an IDD state. Among the cases, 2.8% (n = 31) were in grade one category, while only 0.1% (n = 1) was in grade two category. After 10 years of USI, the goitre prevalence was reduced to 0.1%. Only one case (0.1%) was identified in the 2018 survey (Table 2).

After 10 years of USI implementations in the state, the median UIC had risen significantly from 102.1 µg/L (IQR, 62.3–146.5) to 126.0 µg/L (IQR, 71.2–200.0) (Table 3).

A total of 120 salt samples were analysed in 2008 and 382 salt samples were analysed in 2018. After USI was introduced in Sarawak, the median of iodine content in salt samples was significantly increased from 14.7 mg/kg (IQR, 0.0–27.0 mg/kg) in 2008 to 34.4 mg/kg (IQR, 20.5–43.9 mg/kg) in 2018. In addition, the proportion of salt samples with iodine content less than 20 mg/kg was also decreased significantly from 60.8% in 2008 to 24.6% in 2018 (Table 4).

## 4. Discussion

The incidence of endemic goitre is one of the most widely accepted indicators of the severity of IDD in a region which represents an indicator of long-term nutritional status and the status of median UIC which measurement is more sensitive to current iodine consumption measurement [8,18]. In our study, the findings reveal that the school children iodine status has greatly improved after USI was implemented in Sarawak. In a 2008 study, the total goitre rate (TGR) was found to be very low, indicating that IDD was not a public health issue in Sarawak. In a 2018 study, the TGR was further reduced to 0.1%. This is a promising trend, indicating that the IDD is improving with time. Furthermore, the cassava is no more prominent staple since rice is available all year round [14]. The current rate is much lower than the cut-off points by WHO/UNICEF/ICCIDD classification of 5%. Thus, Sarawak would be classified as the state of no IDD [8]. This finding is in agreement with findings from elsewhere that, once USI has been implemented, the population TGR will be more likely to reduce to less than 5% [20,21,22]. According to Assey [20], after USI was implemented in Tanzania, TGR has decreased from 60.7% in the 1980s to 12.3% in 2004 in the 27 IDD endemic districts. Another study among school-age children in Turkey shows that after 16 years of USI implementation, the goitre rates had significantly decreased from 34.0% in 1999 to 0.3% in 2015 [21]. In 1996, the city of Shanghai implemented USI and the monitoring of goitre prevalence data of children from 1997, 1999, 2005, 2011, 2014, and 2017 revealed that the TGR was at normal levels of 3.07, 0.40, 0.08, 0.08, 0.86, and 1.90%, respectively [22]. This is not surprising because when a population has access to iodized salt, the TGR decreases, after using iodized salt, the TGR pattern changed from severe to mild, and from mild to eliminated IDD, indicating that the population’s iodine levels were adequate [23]. 

The salt iodization programme was considered unsuccessful if IDD still exists in the population and the median UIC remains insufficient. It was categorized as inequitable if median UIC was adequate but IDD were present. It was classified as sustainable when the goal of IDD keeping under control had been achieved [18]. The results of our study showed that after 10 years of USI implemented in Sarawak, the median UIC significantly increased from 102.1 µg/L in 2008 to 126.0 µg/L in 2018. In addition, the distributions of UIC of <50 µg/L before and after USI also showed a significant reduction from 19.2% to 16.6%. Based on the WHO/UNICEF/ICCIDD criteria, our study revealed that the IDD are being kept under control in the population. The improvement of the Sarawak IDD situation in 2018 was likely attributable to the implementation of the mandatory USI programme in 2008 and was in agreement with the results of a number of studies, such as those from Tunisia [24], Iran [25], China [26], and Poland [27] which revealed that the IDD status was significantly improved in the country with iodization programme. A national study was carried out in Tunisia revealed that the median UIC of Tunisian school-aged children was 220 µg/L and only 3.1% of the children had UIC < 50 µg/L [24]. In Iran, the first law requiring mandatory iodization of all salt was passed in 1994 and the surveys from 2003 to 2006 indicated that the median UIC was ≥100 μg/L compared to <100 μgL before the USI programme was implemented [25]. Similarly, a study carried out by Zou et al. [26] in China revealed the median UIC of children aged 8–10 years was 174.3 g/L in the 2013 survey, after the introduction of USI in 1995. Zygmunt et al. [27], with their study on effectiveness of iodine prophylaxis in Poland, found that the median UIC increased continuously from 45.5 μg/L in 1994 (before USI was implemented) to 101.1 μg/L, 100.6 and 288.3 μg/L in 1999, 2010 and 2016, respectively. This is not surprising as the population generally use salt in the preparation of food.

Sarawak’s median salt iodine content has improved from borderline adequate to adequate after 10 years of mandatory USI. When compared to 2008, the proportion of iodized salt with iodine levels of ≥20 mg/kg has increased to from 40% to 75% in 2018. There were several factors that led to this remarkably improvement in the iodized salt quality. The first factor was mainly due to the fact that almost all the iodized salt used in the state were imported. Thus, it is easy for the authority to monitor the compliance of every iodized salt consignment that arrives at any port of entry in Sarawak [12]. The second factor was the strict enforcement of the food legislation at port of entry, distribution, and retail outlets, that a number of them have been charged in court for distributing or selling the iodized salt that contains less or excess of the permitted levels [14]. The last factor was the infrastructure development have made the iodized salt more widely available to communities living rural areas [28]. 

The limitations were that both the 2008 and 2018 studies did not take into consideration the use of iodized salt due to the practicability of measuring the children’s actual daily iodized salt usage. However, the quantity of iodized salt usage was assumed adequate based on the established references [8]. The studies did not measure the data on 24 h dietary recall and intake of supplements which were important in assessment of the effects of food supply on IDD. Another limitation was the palpation of goitre was assessed by several examiners and the result could be skewed by misclassification biases. However, quality control procedures have been adopted through repeated training. Despite these limitations, the studies have many strengths, besides having a high response rate (92.0% and 82.3%). The sampling frame was population-based and included random school children sampling, which can be generalized to the Sarawak population. In addition, the urine samples from children were collected through trained personnel and were analysed in the regional IDD laboratory by experienced medical laboratory technicians. For quality control, 10% of the samples were randomly selected and sent to the national IDD reference laboratory at the Institute for Medical Research in Kuala Lumpur.

## 5. Conclusions

Sarawak has progressed from a state of borderline iodine adequacy (before to the introduction of USI) to one in which the population’s iodine status is now at an ideal level. Findings from this study suggest the effectiveness of mandatory USI in increasing the nutritional iodine status of population in Sarawak. Particular attention is needed to keep the IDD under control in Sarawak, such as improving the monitoring and surveillance system at all levels on the quality of iodized salt to ensure that USI programme is effective in keeping the IDD under control in Sarawak. In addition, In addition, future studies on the prevalence of IDD in other vulnerable groups, such as newborns, pregnant and lactating mothers, is required.

## Figures and Tables

**Table 1 nutrients-14-01585-t001:** Socio-demographic characteristics of the school children.

Characteristics	2008	2018	*p*
	n (%)	n (%)	
Age (Years)			
8	378 (34.2)	315 (31.9)	0.507
9	396 (35.9)	362 (36.7)	
10	330 (29.9)	310 (31.4)	
Gender			
Boys	576 (52.2)	495 (50.1)	0.262
Girls	528 (47.8)	493 (49.9)	
Division			
Southern	569 (51.5)	352 (35.6)	0.001
Central	333 (30.2)	315 (31.9)	
Northern	202 (18.3)	321 (32.5)	
Total	1104	988	

**Table 2 nutrients-14-01585-t002:** Prevalence of goitre before and after USI in Sarawak.

Variables	Goitre Prevalence	
	2008 (N = 1104)	2018 (N = 988)
	n (%)	n (%)
Age (Years)		
8	13 (1.2)	0 (0.0)
9	14 (1.3)	0 (0.0)
10	5 (0.4)	1 (0.1)
Gender		
Boys	16 (1.45)	1 (0.1)
Girls	16 (1.45)	0 (0.))
Division		
Southern	30 (2.7)	0 (0.0)
Central	1 (0.1)	0 (0.0)
Northern	1 (0.1)	1 (0.1)
Total goitre rate (TGR)	32 (2.9)	1(0.1)

**Table 3 nutrients-14-01585-t003:** Median urinary iodine concentrations (UIC) status before and after USI in Sarawak.

Year	n	Urine Iodine (µg/L)		*p*
		Median UIC	IQR	
2008	1104	102.1	62.3–146.5	0.001
2018	988	126.0	71.2–200.0	

**Table 4 nutrients-14-01585-t004:** Salt iodine concentrations (SIC) status before and after USI in Sarawak.

Year	n	Median (IQR)	Salt Iodine Concentrations (mg/kg) Frequency (%)		
			<20	20–39	≥40
2008	120	14.7 (0.0–27.0)	73 (60.8)	38 (31.7)	9 (7.5)
2018	382	34.4 (20.5–43.9)	94 (24.6)	145 (38.0)	143 (37.4)

Mann–Whitney test of median SIC: 2008 vs. 2018, *p* = 0.001. Chi-square test of distribution of SIC: 2008 vs. 2018, *p* = 0.001.

## Data Availability

The data for these studies is not publicly available, but it can be obtained from the Institute for Public Health, National Institutes of Health, Ministry of Health Malaysia, upon reasonable request and with permission from the Director General of Health Malaysia.

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
