# Peer review of "A 10-Year Impact Evaluation of the Universal Salt Iodization (USI) Intervention in Sarawak, Malaysia, 2008–2018"

_nutrients, 2022, doi:10.3390/nu14081585_

Round 1

Reviewer 1 Report

Thank you for an interesting article and an update on the IDD situation in Sarawak.

However, I have an immediate problem in reading the Abstract, in that the baseline data of Goitre Rates (prevalence of 2.9%) and Median Urinary Iodine Concentration of 102 mcg/L, gathered in 2008 are not diagnostic of even mild iodine deficiency in Sarawak at that time. Of course, you show some improvement from 2008 in that there has been a further decrease in goitre prevalence and an increase in MUIC by 2018, but you could not classify Sarawak as being a country with mild iodine deficiency in 2008 and now claim that IDD elimination initiatives by way of introduction of USI have eliminated iodine deficiency in the country.

What is intriguing is that goitre surveys in Sarawak in the 1970s, undertaken first by  Polunin in the early 1970s and later in the 70s by Maberly et al. cited by you, showed a very high prevalence of goitre of almost 100% in some communities, but there was wide geographical and ethnic differences. and large multinodular goitres were very common. Accordingly, two questions immediately arise. First, what strategies have reduced goitre rates and thyroid volumes so dramatically before the introduction of USI in 2008? You mentioned water iodination in some villages as initially being successful, but omitted reference to the original report and the successfully demonstrated system implemented by Maberly et al (Lancet 1981, December 5, 2:1270-1272) in the upper Lemanak River region. When was this initiative discontinued? Second, you make no mention of goitrogens being a cause or contributor to the development of endemic goitre in Sarawak. This reviewer understands that cassava has been an essential food staple in Sarawak and is known to contain cyanogenic glycosides which release thiocyanate that is a well-known goitrogen. You need to address this issue which has been overlooked in your manuscript.

Your current  findings of such low goitre rates are difficult to believe. Why was goitre prevalence less than 5% in 2008 when you say that uptake of iodised salt at the time was unsatisfactory and 50% of children had UIC less than 100 mcg/L? Now you report a goitre prevalence of 0.1% - a level this reviewer finds difficult to accept. Is it that there are virtually no goitres present or is it that your measurement/detection systems for presence of goitre are inadequate? Did you perform any QA studies? Do you have any thyroid ultrasound data to confirm the findings of a goitre prevalence rate of 0.1%? If you do not, you should state in the Discussion that one of the major weaknesses of this study (and probably the one of 2008) has been the reporting of goitre rates by inspection and palpation which is notoriously inaccurate.

With respect to urinary iodine analysis, does your laboratory participate in any external QA/QC programs? This should be addressed in the manuscript.

While your survey was undertaken on representative samples of schoolchildren of appropriate ages, this reviewer feels that you should be careful in declaring that "Sarawak has successfully eliminated IDD in the population". Your data applies only to schoolchildren and complies with WHO/ICCIDD/UNICEF recommendations, but makes no reference to women of childbearing age and in particular, pregnant women. You should make this clear as it is likely that iodine deficiency is prevalent in pregnant women in Sarawak, based upon the data you have gathered. You may have another view, howvere you should discuss this and comment on how you propose to address this issue in the future.
Thank you.

Reviewer 2 Report

Dear colleagues,

Your article summarizes well and clearly the evolution of the IDD prevention programs in Sarawak, Malaysia. It is clear and well written. For your consideration, please see below suggestions to improve it.

Major suggestions:

  1. Introduction: At the end of the first paragraph, you are repeating the statement by WHO of 2003 that 30.6% of the world population have insufficient iodine intake. That is an erroneous estimate because it was based on the proportion of samples below the value of 100 ug I/L, when this criterion is applicable to a median and not to a threshold. It means that if a population has a median equal or larger than 100 ug/L, the whole population was having a sufficient iodine intake. Thus, I suggest replacing that with a more specific value that ou can obtain from reference 4.
  2. Section 3. Results, page 4: Modify the paragraph between tables 2 and 3 because the UIC criteria are based on medians and not in proportion of samples below those medians. Thus, saying that the proportion of school children with UIC of 100-299 (adequate) increased from 4.6 in 2008 to 55.1% in 2018 is not needed. It is sufficient to say that the population median increased from 102.1 ug/L to 125.0 ug/L. Similarly, the proportion of samples above 300 ug/L does not mean that there was an excessive intake of iodine in this population. For making this asseveration, the median of the UIC should have been equal or larger than 300 ug/L, and it was not the case. Similarly, the Table 3 should be simplified leaving only Median and IQR, and if you would like the proportion of samples below <50 ug/L. Another alternative is to present the frequency distribution of UIC of the two years if you would like to show with more details the increment in the iodine contents in the iodized salt.
  3. Section 4. Discussion, page 6: The final section of the paragraph before the section of conclusion. I suggest deleting the section after the reference [28]. You explain as incompliance or because iodine losses the proportion of salt samples that you found with an iodine content below 20 mg I/kg. However, this might be a simple distribution of iodine content around the mean. Therefore, presenting the results of table 4 in a figure of frequency distribution would be appropriate. In other words, the proportion of salt samples with iodine content below 20 or higher than 40 mg I/kg might be a simple distribution (variation) of the iodine content that determined in the salt samples.

 Minor suggestions:

  1. Introduction: Last paragraph in page 1: Improve the grammar of the last sentences. Probably something like this may help: … taking into account several factors: iodine loss from salt is 20% from production site to household, and 20% more during cooking, and average salt intake per capita of 10 g/day, which result in a final estimation of 23 mg I/kg. Therefore, the content of 40 mg I/kg is applicable when the salt intake is slightly higher than 5 g/day.
  2. Introduction: First paragraph of page 2: At the middle of the paragraph, replace the word “continues” with “continued”.
  3. Section 2.2, last paragraph of page 2: Delete a number 8 after the word “criteria” at the middle of the paragraph.
  4. Section 3. Results, page 5: You may present the frequency distribution of the iodine content in the salt for the years 2008 and 2018. This figure has a better meaning that the proportion of iodine contents in the three categories that you used based on the current salt regulation.
  5. Section 4. Discussion, page 5. At the middle of the first paragraph, use the past tense of the verb be: Thus: In 2008 study…. Indicating that IDD was not a public health problem…
  6. Section 4. Discussion, page 5. In the last two lines of the page, delete the number 8 after the word “criteria”. Moreover, I would like to success not to use the word “eliminate” but “keeping under control” because if the salt iodization program deteriorates, the IDD problem will reappear. In other words, IDD are being kept under control and therefore the salt iodization program must continue working well.
  7. Section 5. Conclusion, page 6. Similar to the point above, I would like to suggest to replace the word “eliminating” with “keeping under control”… the menace of IDD in Sarawak.

Round 2

Reviewer 1 Report

Thank you for the significantly improved manuscript. 

In your Introduction you now acknowledge the role of cassava as a goitrogen in the pathogenesis of endemic goitre in Sarawak, however you ignore this issue in the Discussion of your results. If you claim a current goitre prevalence of 0.1% then goitrogens (cassava) must no longer play a role. You must explain this and not ignore the issue as you have done.